# A Method of Determining Multi-Attribute Weights Based on Single-Valued Neutrosophic Numbers and Its Application in TODIM

**Dongsheng Xu [1,2], Yanran Hong [1,]\* and Kaili Xiang [2]**

[1] School of Science, Southwest Petroleum University, Chengdu 610500, China; xudongsheng1976@163.com
[2] School of Economics and Mathematics, Southwestern University of Finance and Economics, Chengdu 611130, China; xiangkl@swufe.edu.cn
\* Correspondence: UranusHYR@163.com

**Abstract:** In this paper, the TODIM method is used to solve the multi-attribute decision-making problem with unknown attribute weight in venture capital, and the decision information is given in the form of single-valued neutrosophic numbers. In order to consider the objectivity and subjectivity of decision-making problems reasonably, the optimal weight is obtained by combining subjective weights and objective weights. Subjective weights are given directly by decision makers. Objective weights are obtained by establishing a weight optimization model with known decision information, then this method will compare with entropy weight method. These simulation results also validate the effectiveness and reasonableness of this proposed method.

**Keywords:** single-valued neutrosophic numbers; determination of weight; TODIM; multi-attribute decision-making; venture capital

## 1. Introduction

Decision-making problems exist everywhere, such as investment decisions [1–4], project evaluation [5–8], scheme optimization [9,10], etc. Research on these issues has been keeping constant, especially in venture capital decision-making. Venture capitalists will do some research on each investment object and judge whether their indicators meet the conditions during investment process. As we all know, risk is uncertain in a broad sense, especially in the future. If all potential risks in the future market can be accurately defined and evaluated, the impact of these risks can be analyzed and the market profits can be maximized. However, the limited cognitive ability of venture capitalists is often not comprehensive enough to understand the detailed information, and makes the obtained decision information unquestionably uncertain and shows some irrationality in data processing. In order to solve the uncertainty, irrationality of object effectively, scholars put forward different theories and methods.

The researches on the existing uncertain decision-making methods mainly focused on the attribute value as language variable, interval number, real number and intuitionistic fuzzy number. Additionally, these decision-making problems with unknown weight were also studied by researchers. It seems that the determination of weight information is the key step of the decision-making problem. Jiang and Wang [11] investigated a method to solve multi-attribute group decision-making (MAGDM) problem with unknown experts weights, where the decision information is given by interval intuitionistic trapezoidal fuzzy numbers. By establishing the optimal model for the decision matrix in the form of the hesitant fuzzy element, Zhang [12] obtained the decision weight and extended it with interval fuzzy value, and finally applied it to the investment selection problem. On the basis of prospect theory

and evidential reasoning approach, Bao et al. [13] proposed an intuitionistic fuzzy decision method and discussed the decision-making problem with unknown attribute weight when the criteria values are given for intuitionistic fuzzy numbers. Based on the interval intuitionistic fuzzy set theory, Xu et al. [14] analyzed the priority of target and gave the calculation method of the weight. Zheng et al. [15] combined triangular fuzzy number with prospect theory to define the foreground value of the attribute distance and defined the association between problem and solution with the grey system theory to get the optimal weights by weight optimization model; Peng and Yang [16] presented two novel interval-valued fuzzy soft set approaches and the combined weight by integrating objective weight with subjective weight. Based on the prospect theory, a novel linguistic decision method with unknown weight under risk is proposed in [17]. Lin et al. [18] studied the multi-attribute decision-making (MADM) problem with unknown attribute weights in the uncertain fuzzy environment. Although these methods can better deal with fuzzy information in decision problems, the portrayal of the uncertainty of attributes is not enough, which leads to the poor integrity and practicality of the decision system. The uncertain information still exists in real life.

Smarandache [19,20] first proposed the idea of neutrosophic set (NS), which includes truth-membership function, indeterminacy-membership function and falsity-membership function, to describe uncertain, discontinuous and incomplete information. In order to simplify NS, Wang et al. [21] defined the single-valued neutrosophic set (SVNS), whose truth-membership, indeterminacy-membership and falsity-membership were represented by single numbers. Then, Wang et al. [22] proposed interval neutrosophic set (INS), whose truth-membership, indeterminacy-membership and falsity-membership were represented by interval numbers. In addition, NS has developed many other forms, such as multi-valued neutrosophic sets (MVNSs), simplified neutrosophic sets (SNSs), complex neutrosophic sets (CNSs) and so on. With a better theoretical foundation, domestic and foreign researchers have applied the theory of neutrosophic sets to MADM in recent years. Considering the MADM problem with the attribute value of SVNN, the ELECTRE method was used to sort the scheme by Peng et al. [23]. Wang et al. [24] put the INS in the TODIM method to seek innovation. Based on the ELECTRE method, Zhang et al. [25] proposed a MADM problem with INS. Liu et al. [26] discussed the ELECTRE method of the interval neutrosophic set. Xu et al. [27] extended the TODIM method for MADM problem with SVNS. An approach is given for the MAGDM problems with SVNNs in [28]. Karasan and Kahraman [29] proposed a novel interval-valued neutrosophic EDAS method for MCDM problem. Xu et al. [30] discussed whether decision information is given by SVNSs or INSs and obtained the corresponding solutions of MADM problems. A method for solving MCGDM problem with SNSs was proposed by Xu et al. [31]. Wang and Zhang [32] combined SVNSs with covering-based rough sets to solve decision-making problem. A new aggregation operators of SVN soft numbers were applied by Chiranjibe and Madhumangal [33] for ranking the alternatives in MADM problems.

These methods are all discussed with the condition of known weights but lack of research on uncertain weights. The ranking results of different weights may have a certain degree of difference, so it needs further study. As mentioned above, the determination of weight information is an important link in the decision-making problem. Aiming at the MADM problem of SVNN with unknown attribute weights, the weight of the index was determined by entropy of SVNN, and the best scheme was selected with the grey correlation analysis (GRA) method based on SVNN Hamming distance in [34]. An entropy measure of SVNS was introduced in [35], and the definitions of distance measure, similarity measure and entropy measure were also proposed in [36], Liu and Luo [37] proposed a new method to calculate similarity measure of the two INSs based on entropy of the INS and finally applied this measure to the medical diagnosis. In [38], Tan et al. determined the attribute weights by using the entropy of neutrosophic sets and extended the VIKOR method to the environment of SVNS for obtaining the emergency group DM method. On the basis of fuzzy exponential entropy, Ye and Cui [39] proposed simplified NSs exponential entropy measures to solve the MADM problem and compared these methods. So it seems that the entropy of NSs can be used to determined the problem with

unknown weight. But there are other method for solution. Zhang and Wu [40] developed a new method to solve single-valued neutrosophic MCDM problem with less detailed weight information. Then, Biswas et al. [41] determined the weight of attributes by maximizing deviations, and sorted them according to grey relational analysis. In order to solve fuzzy MCGDM, Li et al. [42] incorporated power aggregation operators of linguistic neutrosophic numbers with EDAS method(evaluation based on distance from average solution). Peng et al. [43] proposed an MCDM approach with unknown weights to deal with single-valued neutrosophic hesitant fuzzy information. In the single-valued neutrosophic language environment, Ji et al. [44] used the mean-square deviation weighting method to get the criterion weights and select contractors.

Judging from the current research situation, the existing literature mainly uses subjective evaluation methods to estimate weights, but lacks theoretical basis. Under the background of venture capital, the MADM problem with unknown weight is discussed with the two method on the basis of TODIM method and the theory of SVNS. This paper presents the basic theory of single-valued neutrosophic sets and the TODIM method first. By using the entropy of single-valued neutrosoiphic numbers, we can obtain the objective attribute weights. Based on the idea of maximizing deviations, the new method is to establish a weight optimization model for deriving the objective attribute weights. Next, the optimal weights are obtained by weighting the subjective weights and objective weights. Finally, the optimal attribute weights are applied to the TODIM method to solve the MADM problems in venture capital and then rationality and effectiveness of the method are demonstrated.

## 2. Preliminaries

### 2.1. Neutrosophic Sets

**Definition 1** ([20]). *Let X be a space of points(objects), where a generic element in X is denoted by x. A neutrosophic set(NS) A in X is characterized by truth-membership function $T_{A(x)}$, indeterminacy-membership function $I_{A(x)}$ and falsity-membership function $F_{A(x)}$. $T_{A(x)}$, $I_{A(x)}$, $F_{A(x)}$ are the function of finite discrete subset of $[0^-, 1^+]$. It means $T_{A(x)}$, $I_{A(x)}$, $F_{A(x)}$:$X \to [0^-, 1^+]$.*

*So, A can be expressed by $A = \{\langle x, T_A(x), I_A(x), F_A(x) \rangle | x \in X\}$, with the condition of $0^- \leq \sup T_A(x) + \sup I_A(x) + \sup F_A(x) \leq 3^+$.*

**Definition 2.** *$A^C$ is the complement of A, and $T_{A^C} = \{1^+\} \ominus T_A(x)$, $I_{A^C}(x) = \{1^+\} \ominus I_A(x)$, $F_{A^C} = \{1^+\} \ominus F_A(x)$ for every x in X.*

**Definition 3.** *Let A and B be two NSs. $A \subseteq B$ if and only if $\inf T_A(x) \leq \inf T_B(x)$, $\sup T_A(x) \leq \sup T_B(x)$, $\inf I_A(x) \geq \inf I_B(x)$, $\sup I_A(x) \geq \sup I_B(x)$, $\inf F_A(x) \geq \inf F_B(x)$, and $\sup F_A(x) \geq \sup F_B(x)$ for every x in X.*

### 2.2. Single-Valued Neutrosophic Sets

**Definition 4** ([27]). *Let A is a NS, if $T_{A(x)}$:$X \to [0, 1]$, $I_{A(x)} : X \to [0, 1]$, $F_{A(x)} : X \to [0, 1]$, and $0 \leq T_A(x) + I_A(x) + F_A(x) \leq 3$, then A is a single-valued neutrosophic set(SVNS), and is characterized by $A = \{\langle x, T_A(x), I_A(x), F_A(x) \rangle | x \in X\}$.*

*It means that $A = < T_A, I_A, F_A >$, where $T_A \in [0, 1]$, $I_A \in [0, 1]$, $F_A \in [0, 1]$, and $0 \leq T_A + I_A + F_A \leq 3$.*

**Definition 5** ([41]). *$A^C$ is the complement of A and is defined as*

$$A^C = \langle F_A, [1 - I_A], T_A \rangle. \tag{1}$$

**Definition 6** ([41]). *The normalized Euclidean distance between A and B is*

$$d(A, B) = \frac{\sqrt{(T_A - T_B)^2 + (I_A - I_B)^2 + (F_A - F_B)^2}}{3}. \tag{2}$$

**Definition 7** ([24]). *The expectation of A is*

$$E(A) = \frac{(T_A + 1) + (I_A + 1) - F_A}{3}.$$ (3)

**Definition 8** ([33]). *The entropy of A is*

$$En(A) = 1 - \frac{1}{n} \sum_{x_i \in X} (T_A(x_i) + F_A(x_i)) \cdot |I_A(x_i) - I_{A^C}(x_i)|.$$ (4)

## 3. Problem Description

Venture capitalists (VC) should choose the best enterprise (EN) from many investment choices, and then analyze each attribute value of each EN on the basis of known theory, rank these choices and make the best decision. As a rational actor in the market, we need to make appropriate trade-offs in the choice of attributes. These trade-offs can be based on past market experience and industry development rules, and can also be analyzed by human subjective concepts. However, subjective concepts often bring more or less irrational color to risk estimation in venture capital. In order to make the market profitable, it is necessary to minimize the influence of subjective factors on objective reality, especially in the assignment of attribute weights. In the process of subjective analysis, the reliability of the entire estimate will be reduced if the importance of an attribute value is misjudged (that is, the attribute has a higher importance but the given-weight is less). However, if we expand the scope of the analysis as much as possible and obtain more attribute information, it can help the process of analysis to include the uncertainty of future development, but it can also bring great challenges to the reliable estimation, and increases the cost. Therefore, in order to effectively reduce the influence of subjective judgment on each attribute value, we consider the introduction of objective weighting method. Finally, we will combine the subjective and objective opinions to get the optimal weight.

In this paper, we assume that VC wants to invest an EN with his whole funds. At this time, there are $m$ enterprises we can choose for this investment. Before the investment, investors will conduct a survey of $n$ attribute values of various enterprises and then choose the most satisfying enterprises for investment. In brief, the alternatives are $A = (A_1, A_2, \ldots, A_m)$ and the attributes are $G = (G_1, G_2, \ldots, G_n)$. Assume that $a_{ij}$ is the alternative $A_i$, $i = 1, 2, \ldots, m$ under the attribute $G_j$, $j = 1, 2, \ldots, n$.

The characteristics of vc generally meet the following conditions:

(1) In the initial stage of operation, enterprises are generally small and medium-sized enterprises, most of which are high-tech enterprises.
(2) The investment period of a venture capital company is at least three to five years, and the investment mode is equity investment. Overall, these shares account for about 30% of the total shares of the enterprise. However, the investor has no control right, and the EN does not need to provide any guarantee or mortgage to the investor.
(3) Investment must be highly specialized and procedural;
(4) Generally, investors will actively participate in the operation and management of EN to provide value-added services.
(5) The withdrawal of capital through listing, mergers and acquisitions or other forms of equity transfer can enable investors to achieve value-added and excess returns.

The attributes we choose are the most important thing we mentioned before.
Several significant attribute indexes are given here:

(1) Team management.Choosing a good investment project mainly depends on whether the company's team is excellent, especially the leader who leads the team.
(2) Industry prospect.VC firms generally prefer industries with future development potential.

(3)  Competitiveness.  Enterprises with core competitiveness have more advantages than other competitors in the same industry.
(4)  Professional business model, excellent profit model and unique marketing model.
(5)  Both operating income and the proportion of operating profit are high.
(6)  The structures of ownership, top management, enterprises, customers and suppliers are very clear.

## 4. TODIM Method for Single-Valued Neutrosophic Multiple Attribute Decision-Making with Optimal Weight

In this section, we assume that here is a MADM problem and propose the TODIM method with optimal integrated weight to solve it. We assume that $A = (A_1, A_2, \ldots, A_m), (i = 1, 2, \ldots, m)$ are the alternatives, $G = (G_1, G_2, \ldots, G_n), (j = 1, 2, \ldots, n)$ are the attributes, and $w = (w_1, w_2, \ldots, w_n)$ be the weight of $G_j$, where $0 \leq w_j \leq 1$, and $\sum_{j=1}^{n} w_j = 1$. Here, the weight coefficient $w_j (j = 1, 2, \ldots, n)$ is determined by the method mentioned in Section 5. Then we define that $A = (a_{ij})_{m \times n}$ is a decision matrix, where $a_{ij}$ is in the form of SVNN and expresses $A_i$ under $G_j$.

The following shows the complete for the method (Figure 1).

Step 1: The normalized decision matrix is obtained by standardizing the decision information. It means that $A = (a_{ij})_{m \times n}$ will be standardized to get $B = (b_{ij})_{m \times n}$. The efficient factor should not be changed but the cost factor should be changed by its complementary set;

Step 2: Figure out the corresponding weight $w_j (j = 1, 2, \ldots, n)$;

Step 3: The criterion of weight is $w_r = max\{w_j | j = 1, 2, \ldots, n\}$ and the relative weight of $G_j$ to $G_r$ is $w_{jr} = \frac{w_j}{w_r} (j, r = 1, 2, \ldots, n)$;

Step 4:  The dominance degree of $B_i$ over every alternative $B_t$ can be obtained by the following equation

$$\delta(B_i, B_t) = \sum_{j=1}^{n} \varphi_j(B_i, B_t)(i = 1, 2, ..., m),$$ (5)

where

$$\varphi_j(B_i, B_t) = \begin{cases} \sqrt{\frac{w_{jr} d(b_{ij}, b_{tj})}{\sum_{j=1}^{n} w_{jr}}} & , E(b_{ij}) - E(b_{tj}) > 0 \\ 0 & , E(b_{ij}) - E(b_{tj}) = 0 \\ -\frac{1}{\theta} \sqrt{\frac{(\sum_{j=1}^{n} w_{jr}) d(b_{tj}, b_{ij})}{w_{jr}}} & , E(b_{ij}) - E(b_{tj}) < 0 \end{cases},$$ (6)

$$d(b_{ij}, b_{tj}) = \frac{\sqrt{(T_{ij} - T_{tj})^2 + (I_{ij} - I_{tj})^2 + (F_{ij} - F_{tj})^2}}{3},$$ (7)

$$E(b_{ij}) = \frac{(T_{ij} + 1) + (I_{ij} + 1) - (F_{ij})}{3},$$ (8)

In this function, the dominance degree of $B_i$ over $B_t$ under $G_j$ can be represented by $\varphi_j(B_i, B_t)$ and $\theta$ is the loss attenuation coefficient. If $E(b_{ij}) - E(b_{tj}) > 0$, $\varphi_j(B_i, B_t)$ means profits, and if $E(b_{ij}) - E(b_{tj}) < 0$, it means losses;

Step 5: The overall dominance of $B_i$ is given by

$$\xi_i = \frac{\sum_{t=1}^{m} \delta(B_i, B_t) - \min_{1 \leq i \leq m} \{\sum_{t=1}^{m} \delta(B_i, B_t)\}}{\max_{1 \leq i \leq m} \{\sum_{t=1}^{m} \delta(B_i, B_t)\} - \min_{1 \leq i \leq m} \{\sum_{t=1}^{m} \delta(B_i, B_t)\}};$$ (9)

Step 6: According to the value of $\xi_i$ to rank these alternatives, bigger value means better choice.

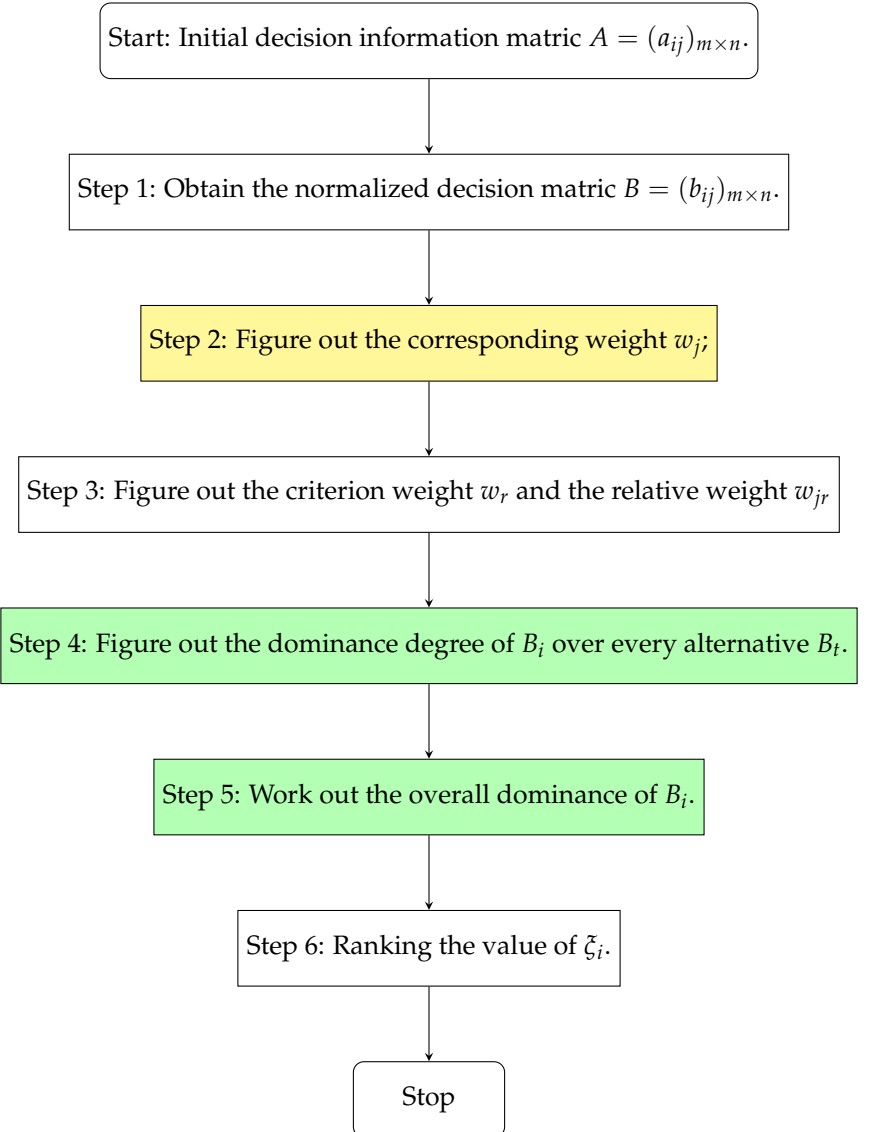

**Figure 1.** TODIM method.

## 5. Determination of Optimal Weight

We assume that the attribute value $a_{ij}$ of the enterprise $A_i$ under $G_j$ is given by single-valued, that is $a_{ij} = <T_{ij}, I_{ij}, F_{ij}>$, $(i = 1, 2, \ldots, m, j = 1, 2, \ldots, n)$, where $T_{ij}$ expresses the truth-membership function of $A_i$ under the attribute $G_j$, $I_{ij}$ expresses the indeterminacy-membership function of $A_i$ under the attribute $G_j$, $F_{ij}$ expresses the falsity-membership function of $A_i$ under the attribute $G_j$.

In this section, we assume that decision makers have assigned the weights $w_j^s$ of these attributes. However, it is not enough and reasonable to use only subjective weight.

### 5.1. Entropy Weight Method

The basic principle of entropy weight method is that greater difference of an index value means smaller the entropy of information, greater information provided by the index, and greater weight of the index. Entropy weight method is based on the real data of the scheme. The result of calculation is more objective and the decision-making result is of high credibility.

In this part, the attribute weight is determined by using the entropy of SVNSs:

$$w_j^e = (1 - En_j) / \sum_{j}^{n} (1 - En_j).\tag{10}$$

Then the optimal weight is given by

$$w_j = \lambda w_j^e + \mu w_j^s.\tag{11}$$

$\lambda, \mu$ are the weighting coefficients of $w_j^e$ and $w_j^s$ respectively.

### 5.2. Weight Optimization Model

In this part, we want to optimize the weight information to make the weight value more reliable. Based on this, this paper assumes that venture capitalists have the best enterprise $R^+$ to invest in and the worst enterprise $R^-$ to reject, in other words, the best enterprise is that each attribute value of the enterprise satisfies the ideal value of investors and on the contrary, it's the worst. So we assume that the ideal value of the best enterprise $R^+$ under the attribute $G_j$ is $R_j^+ = < T_j^+, I_j^+, F_j^+ >$, $j = 1, 2, \ldots, n$, where $T_j^+$ expresses the truth-membership function of $R^+$ under the attribute $G_j$, $I_j^+$ expresses the indeterminacy -membership function of $R^+$ under the attribute $G_j$, $F_j^+$ expresses the falsity -membership function of $R^+$ under the attribute $G_j$.

If $R_j^+$ is an efficient factor,

$$T_j^+ = \max\{T_{ij}\}, I_j^+ = \min\{I_{ij}\}, F_j^+ = \min\{F_{ij}\},\tag{12}$$

and if $R_j^+$ is a cost factor,

$$T_j^+ = \min\{T_{ij}\}, I_j^+ = \max\{I_{ij}\}, F_j^+ = \max\{F_{ij}\},\tag{13}$$

where $i = 1, 2, \ldots, m, j = 1, 2, \ldots, n$.

Based on the idea of deviation maximization, if the greater difference between $A_i$ and $R_j^+$ under the attribute value $G_j$ is, the more important $G_j$ is to sort the solution. So the bigger weight value should be given to the attribute $G_j$. According to this idea, the determination of attribute weight can be given with the following model:

$$\max f(w^+) = w_j^+ \times \sum_{i=1}^{m} \sum_{k=1}^{m} \sum_{j=1}^{n} |d(a_{ij}, R_j^+) - d(a_{kj}, R_j^+)|, \sum_{j=1}^{n} w_j^+ = 1,\tag{14}$$

where

$$d(a_{ij}, R_j^+) = \frac{\sqrt{(T_{ij} - T_j^+)^2 + (I_{ij} - I_j^+)^2 + (F_{ij} - F_j^+)^2}}{3}.\tag{15}$$

By constructing the Lagrange function, the value of $w_j^+$ is

$$w_j^+ = \frac{\sum_{i=1}^{m} \sum_{k=1}^{m} |d(a_{ij}, R_j^+) - d(a_{kj}, R_j^+)|}{\sum_{i=1}^{m} \sum_{k=1}^{m} \sum_{j=1}^{n} |d(a_{ij}, R_j^+) - d(a_{kj}, R_j^+)|}.\tag{16}$$

According to symmetry,

$$w_j^- = \frac{\sum_{i=1}^{m} \sum_{k=1}^{m} |d(a_{ij}, R_j^-) - d(a_{kj}, R_j^-)|}{\sum_{i=1}^{m} \sum_{k=1}^{m} \sum_{j=1}^{n} |d(a_{ij}, R_j^-) - d(a_{kj}, R_j^-)|}.\tag{17}$$

Then, the optimal objective weight is given by the equation:

$$w_j^o = \frac{1}{2}(w_j^+ + w_j^-), \tag{18}$$

Further, we can assume that the optimal weight is given by the equation:

$$w_j = \alpha w_j^o + \beta w_j^s, \ \alpha + \beta = 1. \tag{19}$$

where $\alpha$ and $\beta$ are the weighting coefficients of $w_j^o$ and $w_j^s$, respectively, which are used to measure the proportion of the two and can be selected according to the specific decision problem. It is more convincing to get the corresponding weight information by dealing with the obtained attribute value data.

## 6. Practical Example

In this section, we utilize one numerical examples to illustrate the application of the developed method. All the calculation process is completed with Microsoft Excel.

In order to maximize profits in investment, a VC hopes to choose the best one from many innovative EN for investment. There are three ENs $A = (A_1, A_2, A_3)$ as candidates and three attributes $G = (G_1, G_2, G_3)$. In the end, the final optimal choice between ENs is the object that VC want to choose. We assume that $a_{ij}$ is the alternative $A_i$, $i = 1, 2, 3$ under the attribute $G_j$, $j = 1, 2, 3$. First attribute is cost factor while the next two are efficient factors, so first attribute decision information should be changed by its complementary set and the next two should not be changed. Then we assume that the decision maker gives the decision information in the form of SVNSs, so the decision matric is

$$A = \begin{bmatrix} \langle 0.25, 0.35, 0.35 \rangle & \langle 0.45, 0.45, 0.35 \rangle & \langle 0.55, 0.35, 0.30 \rangle \\ \langle 0.45, 0.25, 0.35 \rangle & \langle 0.50, 0.20, 0.30 \rangle & \langle 0.80, 0.25, 0.45 \rangle \\ \langle 0.75, 0.15, 0.25 \rangle & \langle 0.65, 0.30, 0.20 \rangle & \langle 0.65, 0.35, 0.85 \rangle \end{bmatrix}.$$

Standardizing $A$ to

$$B = \begin{bmatrix} \langle 0.35, 0.65, 0.25 \rangle & \langle 0.45, 0.45, 0.35 \rangle & \langle 0.55, 0.35, 0.30 \rangle \\ \langle 0.35, 0.75, 0.45 \rangle & \langle 0.50, 0.20, 0.30 \rangle & \langle 0.80, 0.25, 0.45 \rangle \\ \langle 0.25, 0.85, 0.75 \rangle & \langle 0.65, 0.30, 0.20 \rangle & \langle 0.65, 0.35, 0.85 \rangle \end{bmatrix}.$$

The expectation matric is

$$E = \begin{bmatrix} E_{11} & E_{12} & E_{13} \\ E_{21} & E_{22} & E_{23} \\ E_{31} & E_{32} & E_{33} \end{bmatrix} = \begin{bmatrix} 0.917 & 0.850 & 0.867 \\ 0.883 & 0.800 & 0.867 \\ 0.783 & 0.917 & 0.717 \end{bmatrix}.$$

where $E_{ij}$ means the expectation of $b_{ij}$.

The distance matrices are

$$D_1 = \begin{bmatrix} d_{11}^1 & d_{12}^1 & d_{13}^1 \\ d_{21}^1 & d_{22}^1 & d_{23}^1 \\ d_{31}^1 & d_{32}^1 & d_{33}^1 \end{bmatrix} = \begin{bmatrix} 0 & 0.075 & 0.183 \\ 0.075 & 0 & 0.111 \\ 0.183 & 0.111 & 0 \end{bmatrix},$$

$$D_2 = \begin{bmatrix} d_{11}^2 & d_{12}^2 & d_{13}^2 \\ d_{21}^2 & d_{22}^2 & d_{23}^2 \\ d_{31}^2 & d_{32}^2 & d_{33}^2 \end{bmatrix} = \begin{bmatrix} 0 & 0.087 & 0.097 \\ 0.087 & 0 & 0.069 \\ 0.097 & 0.069 & 0 \end{bmatrix},$$

$$D_3 = \begin{bmatrix} d_{11}^3 & d_{12}^3 & d_{13}^3 \\ d_{21}^3 & d_{22}^3 & d_{23}^3 \\ d_{31}^3 & d_{32}^3 & d_{33}^3 \end{bmatrix} = \begin{bmatrix} 0 & 0.103 & 0.186 \\ 0.103 & 0 & 0.146 \\ 0.186 & 0.146 & 0 \end{bmatrix}.$$

where $D_j$ denotes the distance matrix $(d_{it}^j)_{3\times3}$ and $d_{it}^j = d(b_{ij}, b_{tj})$ under the attribute $G_j$.

### 6.1. Method 1

We assume that the subjective weights are

$$w_1^s = 0.40, w_2^s = 0.30, w_3^s = 0.30.$$

Then, according to the Equation (4), the entropies of attributes are

$$En_1 = 0.78, En_2 = 0.83, En_3 = 0.81.$$

Next, according to the Equation (10), the attribute weights determined by using the entropy of SVNSs are

$$w_1^e = 0.38, w_2^e = 0.29, w_3^e = 0.33.$$

Apparently, we can see that the value of $w_j^e$ is close to that of $w_j^s$. Finally, according to the Equation (11) and TODIM method, we discuss the effect of different $\lambda$ and $\mu$ on decision result, and the ranking order is shown in Table 1. For simplicity, $A_i$ expresses the overall dominance and $FC$ expresses the final choice of VC.

**Table 1.** Change of $\lambda$ and $\mu$.

| $\lambda$ | $\mu$ | $w_1$ | $w_2$ | $w_3$ | $A_1$ | $A_2$ | $A_3$ | Ranking Order | FC |
|-----------|-------|-------|-------|-------|-------|-------|-------|---------------|-----|
| 0.1 | 0.9 | 0.398 | 0.299 | 0.303 | 1 | 0.604 | 0 | $A_3 < A_2 < A_1$ | $A_1$ |
| 0.3 | 0.7 | 0.394 | 0.297 | 0.309 | 1 | 0.550 | 0 | $A_3 < A_2 < A_1$ | $A_1$ |
| 0.5 | 0.5 | 0.391 | 0.294 | 0.315 | 1 | 0.512 | 0 | $A_3 < A_2 < A_1$ | $A_1$ |
| 0.7 | 0.3 | 0.387 | 0.292 | 0.321 | 1 | 0.484 | 0 | $A_3 < A_2 < A_1$ | $A_1$ |
| 0.9 | 0.1 | 0.383 | 0.290 | 0.327 | 1 | 0.464 | 0 | $A_3 < A_2 < A_1$ | $A_1$ |

As shown in Table 1, the ranking order is $A_3 < A_2 < A_1$ and $A_1$ is the final optimal choice. It is easy to see that different value of $\lambda$ and $\mu$ has no effect on the decision results but has an impact on the overall dominance value of alternative.

### 6.2. Method 2

We also assume that subjective weight is

$$w_1^s = 0.40, w_2^s = 0.30, w_3^s = 0.30.$$

According to the Equations (12) and (13), the attribute value of the best enterprise $R_{1,2,3}^+$ under the attribute $G_{1,2,3}$ is

$$R_1^+ = \langle 0.25, 0.35, 0.35 \rangle, R_2^+ = \langle 0.65, 0.20, 0.20 \rangle, R_3^+ = \langle 0.80, 0.25, 0.30 \rangle.$$

and the attribute value of the worst enterprises $R_{1,2,3}^-$ under the attribute $G_{1,2,3}$ is

$$R_1^- = \langle 0.75, 0.15, 0.25 \rangle, R_2^- = \langle 0.45, 0.45, 0.35 \rangle, R_3^- = \langle 0.55, 0.35, 0.85 \rangle.$$

According to the Equations (16) and (17),

$$w_1^+ = 0.311, \quad w_2^+ = 0.256, \quad w_3^+ = 0.433,$$
$$w_1^- = 0.324, \quad w_2^- = 0.266, \quad w_3^- = 0.411.$$

and according to the Equation (18), the optimal objective weight is

$$w_1^o = 0.317, w_2^o = 0.261, w_3^o = 0.422.$$

Apparently, we can see that the value of $w_j^o$ is somewhat deviated from that of $w_j^s$. Next, according to the Equation (11) and TODIM method, we discuss the effect of different weight parameters $\alpha$ and $\beta$ on the decision result, and the ranking order is shown in Table 2. Similarly, $A_i$ expresses the overall dominance and *FC* expresses the final choice of VC.

**Table 2.** Change of $\alpha$ and $\beta$.

| $\alpha$ | $\beta$ | $w_1$ | $w_2$ | $w_3$ | $A_1$ | $A_2$ | $A_3$ | Ranking Order | FC |
|------|------|-------|-------|-------|-------|-------|-------|---------------|-----|
| 0.1 | 0.9 | 0.572 | 0.296 | 0.132 | 1 | 0.605 | 0 | $A_3 < A_2 < A_1$ | $A_1$ |
| 0.3 | 0.7 | 0.515 | 0.288 | 0.196 | 1 | 0.552 | 0 | $A_3 < A_2 < A_1$ | $A_1$ |
| 0.5 | 0.5 | 0.459 | 0.280 | 0.261 | 1 | 0.516 | 0 | $A_3 < A_2 < A_1$ | $A_1$ |
| 0.7 | 0.3 | 0.402 | 0.273 | 0.325 | 1 | 0.489 | 0 | $A_3 < A_2 < A_1$ | $A_1$ |
| 0.9 | 0.1 | 0.346 | 0.265 | 0.389 | 1 | 0.471 | 0 | $A_3 < A_2 < A_1$ | $A_1$ |

As shown in Table 2, the ranking order is $A_3 < A_2 < A_1$ and $A_1$ is the final optimal choice. It is also easy to see that the different value of $\alpha$ and $\beta$ has no effect on the decision results but has an impact on the overall dominance value of alternative.

*6.3. Analysis*

From Section 6.1, it can be seen that the objective weights we get from Section 5.1 are roughly same with the subjective weights. Also from Section 6.2, the objective weights we get from Section 5.2 are quite different form the subjective weights. This shows that even if the decision information is the same, different objective weights can be obtained by different weight determination methods. However, we can see that the two methods have the same result. It means that the new method is feasible, but the combination of subjective and objective in this new method is more reasonable and convincing. The insufficient number of attribute indicators and candidate enterprises discussed in this paper may lead to some shortcomings in the results, which need to be further explored in future research.

**7. Conclusions**

In order to solve MADM problems with unknown weights in venture capital, TODIM method is used to choose the best enterprise to invest in this paper. We assume that the subjective weights of decision-maker have been given in this paper firstly, and then combine it with the objective weights. The objective weights are obtained by the weight optimization model on the basis of the idea of deviation maximization. Then, we compare this new method with the common entropy weight method for empirical analysis. It can be easy to see that the value of $w_j^e$ is close to that of $w_j^s$ while the value of $w_j^o$ is somewhat deviated from that of $w_j^s$. This shows that even if the decision information is the same, different objective weights can be obtained by different weight determination methods. However, we can find that the results of the two methods are consistent by comparing the results, so the feasibility and rationality of the new method are demonstrated. The two methods reduce the impact of subjectivity and objectivity on decision-making and the result is more credible. The insufficient number of attribute indicators and candidate enterprises discussed in this paper may lead to some shortcomings in the results, which need to be further explored in future research.

**Author Contributions:** Formal analysis, K.X.; Methodology, Y.H.; Validation, Y.H.; Writing—original draft, Y.H.; Writing—review & editing, D.X.

**Funding:** This research was funded by the Humanities and Social Sciences Foundation of Ministry of Education of the Peoples Republic of China (17YJA630115).

**Conflicts of Interest:** The authors declare no conflict of interest.

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
