# Peer review of "A Method of Determining Multi-Attribute Weights Based on Single-Valued Neutrosophic Numbers and Its Application in TODIM"

_symmetry, doi:10.3390/sym11040506_

Round 1

Reviewer 1 Report

Please find attached the review report.

Author Response

Dear Reviewer,

Thank you very much for your suggestions.

Reviewer 2 Report

Review for manuscript ID symmetry-475520

a)            Positive aspects:

As theme, methodology and as value, this article is adapted to journal „Symmetry”. The thesis of the manuscript is clear. This article shows a very good knowledge of single-valued neutrosophic numbers. The research central results are relevant and verifiable. The calculations are correct.

b)            Negative aspects:

b1) This manuscript does not prove a very good knowledge of TODIM method, in that it does not explain the foundation contributions of the method. As such, this aspect needs to be developed.

b2) This manuscript does not show openness for the European bibliography in relation to the topic, the key words. Almost European research is not taken into account. Subsequently, it is necessary to complete the bibliography on the above mentioned idea.

c) Conclusion: This manuscript requires major corrections.

Author Response

(The authors gave the same response as above.)

Round 2

Reviewer 1 Report

The remarks regarding the contents are dealt with in a satisfactory way, as such the paper can be accepted for publication in Symmetry.

Reviewer 2 Report

I thank the author/authors for the elegance of showing respect for a collegial, objective and good-faith point of view.

In the revised form, the article fits into the scientific value standards of "Symmetry".

I agree with the publication in the current form.